# Transitions of Liver and Biliary Enzymes during Proton Beam Therapy for Hepatocellular Carcinoma

**DOI:** 10.3390/cancers12071840

**Published:** 2020-07-08

**Authors:** Taisuke Sumiya, Masashi Mizumoto, Yoshiko Oshiro, Keiichiro Baba, Motohiro Murakami, Shosei Shimizu, Masatoshi Nakamura, Yuichi Hiroshima, Toshiki Ishida, Takashi Iizumi, Takashi Saito, Haruko Numajiri, Kei Nakai, Toshiyuki Okumura, Hideyuki Sakurai

**Affiliations:** 1Department of Radiation Oncology, Proton Medical Research Center, University of Tsukuba Hospital, Tsukuba 305-8576, Japan; sumiya.taisuke@gmail.com (T.S.); ooyoshiko@hotmail.com (Y.O.); baba@pmrc.tsukuba.ac.jp (K.B.); murakami@pmrc.tsukuba.ac.jp (M.M.); shimizu@pmrc.tsukuba.ac.jp (S.S.); nakamura@pmrc.tsukuba.ac.jp (M.N.); hiroshima@pmrc.tsukuba.ac.jp (Y.H.); tishida@pmrc.tsukuba.ac.jp (T.I.); iizumi@pmrc.tsukuba.ac.jp (T.I.); saitoh@pmrc.tsukuba.ac.jp (T.S.); haruko@pmrc.tsukuba.ac.jp (H.N.); knakai@pmrc.tsukuba.ac.jp (K.N.); okumura@pmrc.tsukuba.ac.jp (T.O.); hsakurai@pmrc.tsukuba.ac.jp (H.S.); 2Department of Radiation Oncology, Tsukuba Medical Center Hospital, Tsukuba 305-8558, Japan

**Keywords:** radiotherapy, proton beam therapy, hepatocellular carcinoma, liver, liver and biliary enzymes, bilirubin

## Abstract

Proton beam therapy (PBT) is a curative treatment for hepatocellular carcinoma (HCC), because it can preserve liver function due to dose targeting via the Bragg peak. However, the degree of direct liver damage by PBT is unclear. In this study, we retrospectively analyzed liver/biliary enzymes and total bilirubin (T-Bil) as markers of direct liver damage during and early after PBT in 300 patients. The levels of these enzymes and bilirubin were almost stable throughout the treatment period. In patients with normal pretreatment levels, aminotransferase (AST), alanine aminotransferase (ALT), alkaline phosphatase (ALP), gamma-glutamyl transpeptidase (GGT), and T-Bil were abnormally elevated in only 2 (1.2%), 1 (0.4%), 0, 2 (1.2%), and 8 (3.5%) patients, respectively, and in 8 of these 13 patients (61.5%) the elevations were temporary. In patients with abnormal pretreatment levels, the levels tended to decrease during PBT. GGT and T-Bil were elevated by 1.62 and 1.57 times in patients who received 66 Gy (RBE) in 10 fractions and 74 Gy (RBE) in 37 fractions, respectively, but again these changes were temporary. These results suggest that direct damage to normal liver caused by PBT is minimal, even if a patient has abnormal pretreatment enzyme levels.

## 1. Introduction

Hepatocellular carcinoma (HCC) is the most common cancer in the liver, accounting for approximately 90% of primary liver cancers [1,2,3]. Surgery is the first treatment choice, but highly accurate radiotherapy has also been used in recent years [4,5,6]. As one modality of radiotherapy, we have obtained many favorable results of proton beam therapy (PBT) for HCC [7,8,9,10,11,12,13,14,15,16,17,18,19,20,21,22], including good preservation of liver function after PBT [21]. However, liver function defined by the Child–Pugh score and indocyanine green retention rate at 15 min (ICG-R15) can be affected by tumor progression or liver cirrhosis, and this can make direct liver damage due to radiotherapy unclear. Therefore, in this study, we examined direct liver damage based on analysis of hepatobiliary enzymes and bilirubin, as markers of damage to hepatocytes and cholangiocytes [23,24,25,26,27,28].

## 2. Results

### 2.1. Patients Characteristics

The patients’ characteristics are shown in Table 1. The patients included 219 males and 81 females, and had a median age of 73 (range 27 to 91) years old. The Child classification was A in 237 patients (Pugh scores of 5 in 165 and 6 in 72 patients), B in 62 patients (Pugh scores of 7 in 36, 8 in 19, and 9 in 7 patients), and C in 1 patient (Pugh score of 10). Hepatitis B and C were present in 51 and 140 patients, respectively, and 44 patients had alcoholic liver cirrhosis. Portal vein tumor thrombosis (PVTT) was observed in 48 patients: Vp1 in 3, Vp2 in 16, Vp3 18, and Vp4 in 11. The three PBT protocols of 66 Gy (relative biological effectiveness (RBE)) in 10 fractions (10fr group), 72.6 Gy (RBE) in 22 fractions (22fr group), and 74 Gy (RBE) in 37 fractions (37fr group) were administered in 77 (26%), 148 (50%), and 75 (20%) patients, respectively. All patients received no additional therapy for HCC within 3 months after completion of PBT.

The only differences in patient characteristics and pretreatment laboratory data among the three PBT protocols were for PVTT, biliary enzymes (alkaline phosphatase (ALP) and gamma-glutamyl transpeptidase (GGT)), and clinical target volume (CTV) in the 10fr group. There were no PVTT cases in this group (0% vs. 25% and 14.7% in the 22fr and 37fr groups, respectively, *p* = 0.000). Biliary enzymes were lower in the 10fr group than in the other groups, which may have been due to the tumor location and absence of PVTT. CTV was significantly smaller in the 10fr group (23.5 vs. 93 and 113 cc in the 22fr and 37fr groups, respectively, *p* < 0.001).

### 2.2. Transitions of the Laboratory Data

The levels of enzymes and total bilirubin (T-Bil) were almost stable throughout the treatment period, based on Welch analysis. Changes of these enzymes are shown in Figure 1, Figure 2 and Figure 3 for the three treatment protocols. Some enzymes decreased after PBT. In the 22fr group, alanine aminotransferase (ALT) was significantly lower at 5 weeks and 2 months compared to the pretreatment level (27.0 and 26.1 vs. 32.1 U/l, *p* = 0.0477, *p* = 0.0145, respectively), and GGT was significantly lower at 3 months after PBT (61.2 vs. 100.0 U/l, *p* = 0.0107). In the 10fr group, T-Bil was significantly lower at 1 month after PBT compared to pretreatment (0.7 vs. 1.0 mg/dl, *p* = 0.0173).

Pretreatment aspartate aminotransferase (AST), ALT, ALP, GGT, and T-Bil had normal values in 163, 233, 196, 171, and 227 patients, respectively. In these patients, values were abnormally elevated (>Common Terminology Criteria for Adverse Events (CTCAE) grade2) during the treatment period in 2, 1, 0, 2, and 8 patients, respectively (Table 2). Among these 13 patients, temporary elevation of ALT and GGT occurred in 8 patients, and all these changes recovered. AST elevation occurred in 2 patients in the 37fr group, and declined in one of these patients. However, AST remained elevated in one patient, and tumor progression out of the treatment field was observed in this case. Elevation of T-Bil was observed in 8 patients in the 22fr group and persisted for over 3 months in 3 patients, for unclear reasons. The significant risk factors for elevation were pretreatment liver function for T-Bil (*p* < 0.001), and CTV for ALP (*p* = 0.0243).

Abnormal pretreatment values of AST, ALT, ALP, GGT, and T-Bil were present in 137, 67, 104, 129, and 73 patients, respectively. The range of abnormal values was wide, so changes were evaluated as percentages. The median value was within −35.9% to +62.1% for all enzymes, and most medians were within −30% to +10% (Figure 4). In patients in the 10fr group, GGT at 3 weeks was elevated to +62.1% but rapidly decreased to −26.2% at 1 month after PBT (7 weeks from the start of PBT), and T-Bil was elevated to +57.3% at 2 months after PBT but decreased to +13.3% at 3 months after PBT.

Pretreatment and post treatment Child–Pugh score and ALBI score were also investigated. The median Child–Pugh scores were 5 (range: 5 to 10) at pretreatment period and 6 (range: 5 to 10) at the end of the treatment. The median ALBI scores were −2.43 (range: −3.50 to −0.73) at pretreatment period and −2.42 (range: −3.36 to 0.04) at the end of the treatment, respectively. There were no significant changes between pre- and post-treatment (*p* = 0.511 and *p* = 0.199, respectively).

### 2.3. Case Presentation

As a typical example, we present the case of a 64-year-old male patient who received PBT at 72.6 Gy (RBE) in 22 fractions. The maximum tumor size was 18 cm in diameter, and the CTV was 3236 cc. Figure 5 shows the treatment planning. The tumor was large and the patch technique [10] was used for this patient, but spreading of the low dose area was minimized. Changes in laboratory data for the patient are shown in Figure 6. The normal pretreatment T-Bil of 1.2 mg/dl elevated to 2.1 mg/dl at 1 week after the start of PBT, and ursodeoxycholic acid was prescribed. T-Bil then declined to 1.2 mg/dl at 2 months after PBT. The patient survived for 2 years after PBT, but then died due to out-of-field recurrence.

## 3. Discussion

Radiotherapy techniques have progressed in recent decades, and radiotherapy has been used for radical treatment of HCC for patients who are not indicated for surgical resection or ablative therapy. The proton beam used in PBT has a narrow energy peak called the Bragg peak that can reduce the dose area compared to photon beam therapy, such as stereotactic body radiotherapy and intensity modulated radiotherapy [29,30,31,32]. Therefore, the dose distribution in PBT is favorable for HCC, and PBT is widely indicated for HCC, including in patients with large HCC [10,15] and PVTT [9,14]. The results of PBT for HCC have been favorable, with liver function defined by Child–Pugh score or ICG-R15 well preserved in these reports [20,21]. However, these measures of liver functions are affected by tumor progression or liver cirrhosis. Therefore, to obtain a direct measure of liver damage by PBT, we examined levels of liver/biliary enzymes and T-Bil in the current study.

AST, ALT, and ALP levels are commonly used to evaluate radiation-induced liver disease (RILD) [33]. These enzymes are abundant in hepatocytes and are released when hepatocytes are damaged. ALT is more specific for liver due to its almost exclusive presence in hepatocytes and nephrocytes, whereas AST is also found in the heart and muscle [23,24,25]. ALP is found on the canalicular membrane in hepatocytes, and its blood level reflects biliary duct obstruction (BDO). ALP can be used to detect a micro-obstruction without an increase of bilirubin, but interpretation of a high ALP level can be difficult because ALP is also present in bone, placenta, intestine, and kidney. GGT, another enzyme in the canalicular membrane, is not found in bone, and thus elevations of both ALP and GGT suggest an effect in liver [23,26]. Bilirubin is a metabolite of old erythrocytes that is excreted in bile after conjugation in the liver and reflects hepatocyte dysfunction or BDO [23,27].

There are a few reports of laboratory data, including these enzymes, related to PBT. In a phase II study of PBT in 76 patients with HCC, Bush et al. found no significant changes in posttreatment AST, ALT, ALP, bilirubin, and albumin levels [34]. In repeated PBT for HCC, Oshiro et al. found no significant elevation of AST, ALT, and platelet count between the first and second PBT [22]. These studies suggest that late elevation of these enzymes and late liver toxicities are minor in PBT.

In this study, liver enzyme levels were almost stable throughout the treatment period, and elevations of enzymes were almost always temporary. In fact, liver and biliary enzymes decreased in some patients, especially in those with tumors at the porta hepatis. This may be due to recovery of hepatocytes or cholestasis through improvement of liver damage caused by the tumor. The CTV was not a significant factor associated with a change of liver enzymes. Generally, the lower dose area as a target volume is extended in photon radiotherapy. In an analysis of the functional liver volume detected by single photon emission computed tomography (SPECT), Shirai et al. observed a dysfunctional liver area induced by photon radiotherapy, with an increased risk of RILD in cases in which a high percentage of the functional liver area was destroyed by the treatment [35]. In contrast, PBT can make a smaller irradiated area due to the Bragg peak. This is one of the reasons that the target volume does not affect laboratory data for liver enzymes.

There are some limitations in this study, including the retrospective design and possible selection bias. Additionally, we do not compare PBT with photon beam therapy; therefore, the actual benefits of PBT are unclear. However, we found that liver and biliary enzymes were almost stable during and early after PBT, and some patients with high enzyme or bilirubin levels had decreased levels after PBT. These results suggest that PBT-induced direct damage of normal liver was minimal.

## 4. Materials and Methods

### 4.1. Patients

PBT was conducted for 1063 HCC lesions between August 2007 and October 2019 at our hospital. Of these cases, 300 patients who received protocol treatment and met the following criteria was retrospectively analyzed (Tsukuba Clinical Research & Development Organization R01-146): (1) no active lesions out of the target volume, (2) no metastases beyond the liver, (3) no other radiotherapy (including the liver before the treatment), (4) all treatments in the same period and using the same protocol if multiple lesions were treated, (5) laboratory data for more than one time point during treatment, and (6) no unexpected interruptions of treatment. Levels of liver/biliary enzymes, including AST, ALT, ALP, GGT, and T-Bil, were evaluated before and up to three months after PBT.

### 4.2. Proton Beam Therapy

Fiducial markers (iridium seeds of 2 mm in length and 0.8 mm in diameter) were implanted adjacent to the tumor prior to treatment planning. After immobilization of the patient, planning CT images were obtained at 2.5-mm intervals during the expiratory phase using a respiratory gating system (Anzai Medial Co., Tokyo, Japan) [36,37]. The CTV was defined as the gross tumor volume (GTV) plus a 5- to 10-mm margin in all directions. In a case with a PVTT, the margin was extended to ≥10 mm. The internal target volume (ITV) was defined as the CTV plus an additional 5-mm margin on the caudal axes to compensate for the uncertainty of respiratory movement. Proton beams from 155 to 250 MeV generated through a linear accelerator and synchrotron were spread out and shaped with ridge filters, double scattering sheets, multicolimators, and a custom-made bolus to ensure that the beams conformed to the planning data.

We selected the fractionations by the tumor locations. When the tumors were adjacent to high-risk organs, the fraction doses were reduced to avoid severe chronic adverse effects: 2Gy (RBE) per fraction for the tumors near the gastrointestinal tract to prevent gastrointestinal hemorrhage, 3.3Gy (RBE) per fraction for the tumors near the porta hepatis to prevent severe bleeding and biliary tract obstruction, and 6.6Gy (RBE) for the others. Then, treatment schedules were established as 74 Gy (RBE) in 37 fractions for the tumors within 2 cm of the gastrointestinal tract, 72.6 Gy (RBE) in 22 fractions for the tumors within 2 cm of the porta hepatis, and 66 Gy (RBE) in 10 fractions for all the other tumors in our institution. The gastrointestinal tract was avoided as far as possible after 40 to 50 Gy (RBE) irradiated. Previous study has suggested that these three fractionations can achieve similar therapeutic effects [13]. These doses were calculated assuming the RBE to be 1.1.

### 4.3. Follow-Up Procedures

During treatment, acute treatment-related toxicities and treatment efficacy were assessed once or twice per week in all patients. After completion of PBT, physical examinations, blood sampling, and CT or MRI were performed at intervals of several months. If it was difficult for patients to visit our hospital due to traveling distance or a poor general condition, follow up was performed at a nearby hospital and the results were sent to our hospital.

### 4.4. Statistical Analysis

Changes of hepatic/biliary enzymes and T-Bil levels were analyzed for patients with each treatment protocol, because the treatment period differed among the protocols. Differences of patient characteristics among protocols were evaluated by Pearson chi-square test or Fisher exact test [38]. A Welch t-test was used to evaluate weekly changes in laboratory data from pretreatment data [39]. Laboratory data were also classified based on CTCAE v.5.0. All analyses were performed using EZR (http://www.jichi.ac.jp/saitama-sct/SaitamaHP.files/statmed.html), which is based on R (R Foundation for Statistical Computing, Vienna, Austria) [40].

## 5. Conclusions

Our result showed that direct damage to normal liver caused by PBT is minimal and temporary. In addition, the levels of liver and biliary enzymes or bilirubin tend to decrease in the patients with abnormal pretreatment levels.

## Figures and Tables

**Figure 1 cancers-12-01840-f001:**
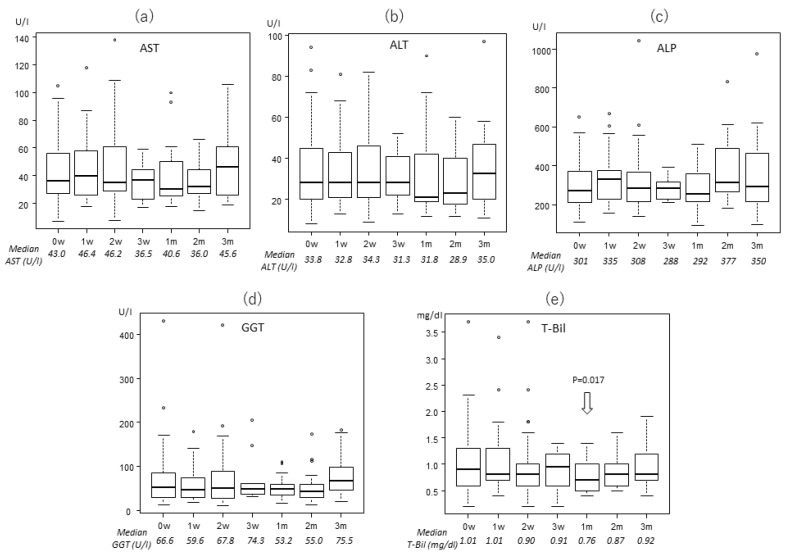
Changes in laboratory data for patients who received 66 Gy (RBE) in 10 fractions: (**a**) aminotransferase (AST), (**b**) aminotransferase (ALT), (**c**) alkaline phosphatase (ALP), (**d**) gamma-glutamyl transpeptidase (GGT), and (**e**) total bilirubin (T-Bil).

**Figure 2 cancers-12-01840-f002:**
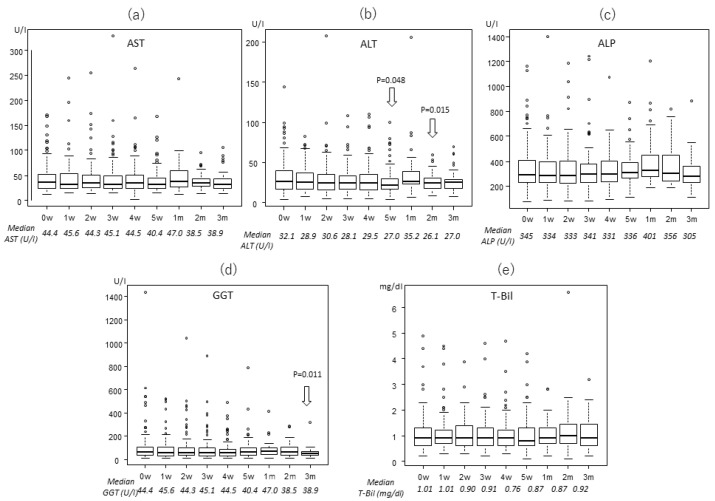
Changes in laboratory data for patients who received 72.6 Gy (RBE) in 22 fractions: (**a**) AST, (**b**) ALT, (**c**) ALP, (**d**) GGT, and (**e**) T-Bil.

**Figure 3 cancers-12-01840-f003:**
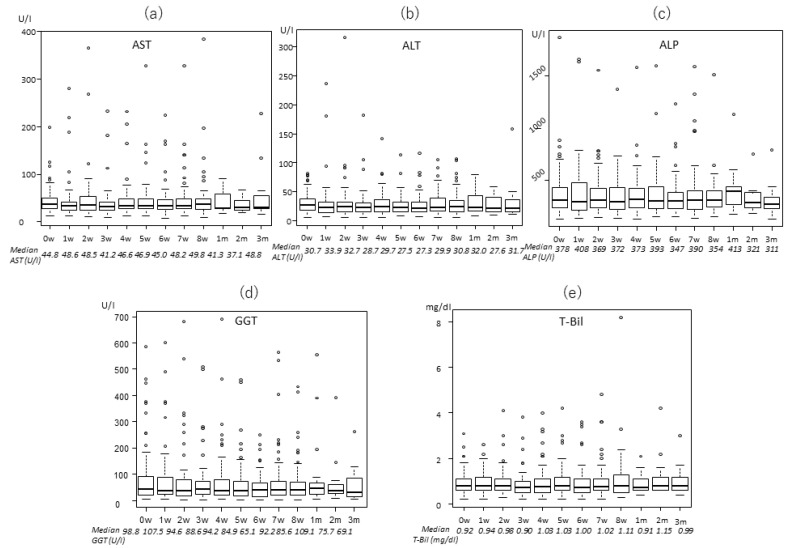
Changes in laboratory data for patients who received 74 Gy (RBE) in 37 fractions: (**a**) AST, (**b**) ALT, (**c**) ALP, (**d**) GGT, and (**e**) T-Bil.

**Figure 4 cancers-12-01840-f004:**
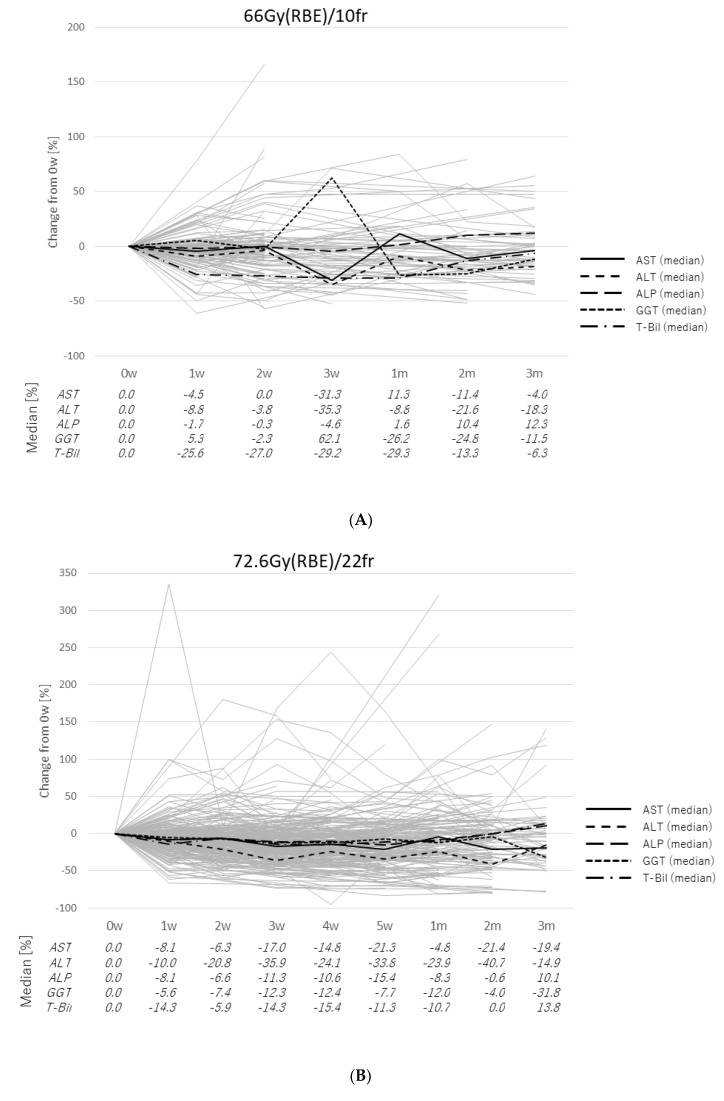
(**A**) Changes in laboratory data for patients with abnormal pretreatment values treated by 66Gy (RBE) in 10 fractions. (**B**) Changes in laboratory data for patients with abnormal pretreatment values treated by 72.6Gy (RBE) in 22 fractions. (**C**) Changes in laboratory data for patients with abnormal pretreatment values treated by 74Gy (RBE) in 37 fractions.

**Figure 5 cancers-12-01840-f005:**
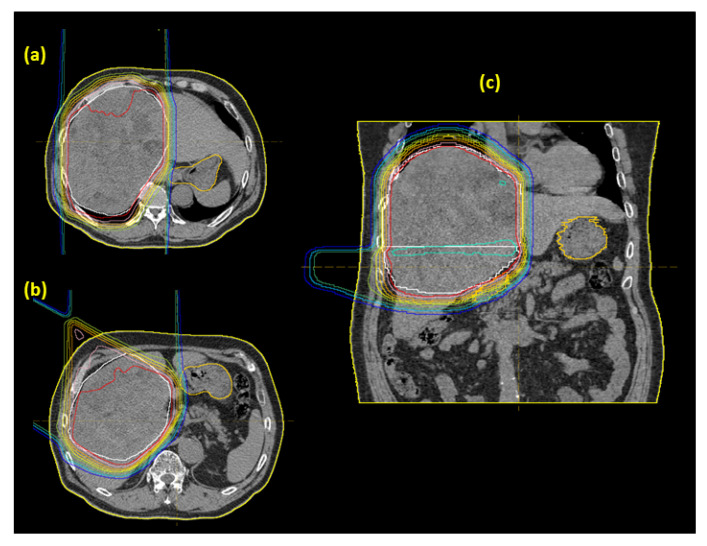
Dose distribution in a 64-year-old patient with a large tumor that was treated using the patch technique: (**a**) upper treatment field, (**b**) lower treatment field, and (**c**) coronal view of the merged fields.

**Figure 6 cancers-12-01840-f006:**
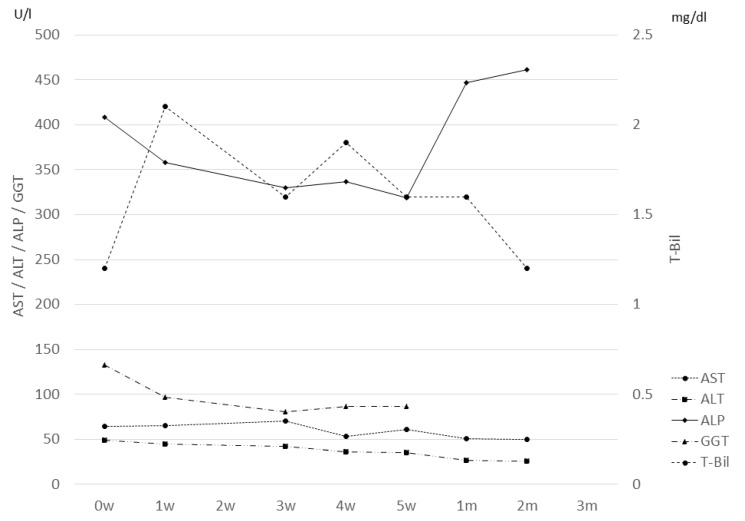
Changes in laboratory data for a 64-year-old patient with a large tumor.

**Table 1 cancers-12-01840-t001:** Characteristics of patients and tumors.

Proton Therapy	Total	66 Gy (RBE)/10 Fractions	72.6 Gy (RBE)/22 Fractions	74 Gy (RBE)/37 Fractions
Patient number (%)	300	77 (25.7%)	148 (49.3%)	75 (25%)
Median age (range)	73 (27–91)	70 (42–88)	72.5 (33–91)	77 (27–91)
Gender				
Male	219 (73%)	53 (68.8%)	114 (77%)	52 (69.3%)
Female	81 (27%)	24 (31.2%)	34 (23%)	23 (30.7%)
Alcoholic liver cirrhosis				
No	256 (85.3%)	69 (89.6%	121 (81.8%)	66 (88%)
Yes	44 (14.7%)	8 (10.4%)	27 (18.2%)	9 (12%)
Hepatitis B				
No	249 (83%)	65 (84.4%)	120 (81.1%)	64 (85.3%)
Yes	51 (17%)	12 (15.6%)	28 (18.9%)	11 (14.7%)
Hepatitis C				
No	160 (53.3%)	35 (45.5%)	86 (58.1%)	39 (52%)
Yes	140 (46.7%)	42 (54.5%)	62 (41.9%)	36 (48%)
Pugh score				
5	165 (55%)	48 (62.3%)	79 (53.4%)	38 (50.7%)
6	72 (24%)	16 (20.8%)	35 (23.6%)	21 (28%)
7	36 (12%)	7 (9.1%)	17 (11.5%)	12 (16%)
8	19 (6.3%	5 (6.5%)	12 (8.1%)	2 (2.7%)
9	7 (2.3%)	1 (1.3%)	4 (2.7%)	2 (2.7%)
10	1 (0.3%)	0 (0%)	1 (0.7%)	0 (0%)
Portal vein thrombosis (Vp)				
0	252 (84%)	77 (100%)	111 (75%)	64 (85.3%)
1	3 (1%)	0 (0%)	3 (2%)	0 (0%)
2	16 (5.3%)	0 (0%)	12 (8.1%)	4 (5.3%)
3	18 (6%)	0 (0%)	13 (8.8%)	5 (6.7%)
4	11 (3.7%)	0 (0%)	9 (6.1%)	2 (2.7%)
Median CTV (cc)	84.5 (3–3236)	23.5 (7–337)	93 (3–3236)	113 (6–1681)
Median pretreatment values (range)				
AST (U/l)	36 (7–199)	36 (7–105)	35.5 (13–171)	37 (13–199)
ALT (U/l)	27 (4–145)	28 (8–94)	26 (4–145)	26 (6–82)
ALP (U/l)	294 (78–1869)	269 (110–652)	292.5 (78–1162)	308 (129–1869)
GGT (U/l)	55 (13–1435)	51 (13–431)	59 (13–1435)	54 (16–599)
T-Bil (mg/dl)	0.8 (0.2–4.9)	0.9 (0.2–3.7)	0.9 (0.2–4.9)	0.8 (0.2–3.1)

**Table 2 cancers-12-01840-t002:** Maximum grade of enzymes for patients with normal pretreatment values.

Laboratory Data	CTCAE Grade	Fractionation of PBT			*p*-Value	Number of Patients
		10fr	22fr	37fr		
AST	0	32 (80%)	68 (81.9%)	27 (67.5%)	0.204	127 (77.9%)
	1	8 (20%)	15 (18.1%)	11 (27.5%)		34 (20.9%)
	2	0 (0%)	0 (0%)	1 (2.5%)		1 (0.6%)
	3	0 (0%)	0 (0%)	1 (2.5%)		1 (0.6%)
	Total	40	83	40		163
ALT	0	51 (89.5%)	102 (87.9%)	49 (81.7%)	0.429	202 (86.7%)
	1	6 (10.5%)	14 (12.1%)	10 (16.7%)		30 (12.9%)
	2	0 (0%)	0 (0%)	1 (1.7%)		1 (0.4%)
	Total	57	116	60		233
ALP	0	42 (76.4%)	73 (76.8%)	32 (69.6%)	0.636	147 (75%)
	1	13 (23.6%)	22 (23.2%)	14 (30.4%)		49 (25%)
	Total	55	95	46		196
GGT	0	44 (91.7%)	65 (79.3%)	36 (87.8%)	0.130	145 (84.8%)
	1	3 (6.2%)	16 (19.5%)	5 (12.2%)		24 (14%)
	2	1 (2.1%)	0 (0%)	0 (0%)		1 (0.6%)
	3	0 (0%)	1 (1.2%)	0 (0%)		1 (0.6%)
	Total	48	82	41		171
T-Bil	0	50 (86.2%)	85 (78%)	49 (81.7%)	0.105	184 (81.1%)
	1	8 (13.8%)	17 (15.6%)	10 (16.7%)		35 (15.4%)
	2	0 (0%)	7 (6.4%)	0 (0%)		7 (3.1%)
	3	0 (0%)	0 (0%)	1 (1.7%)		1 (0.4%)
	Total	58	109	60		227

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
