# Peer review of "Transitions of Liver and Biliary Enzymes during Proton Beam Therapy for Hepatocellular Carcinoma"

_cancers, 2020, doi:10.3390/cancers12071840_

Round 1

Reviewer 1 Report

Sumiya T, et al. reported transitions of liver and biliary enzymes during proton beam therapy for hepatocellular carcinoma. The concept of this study seems to be interesting and clinically important. However, there are some concerns that should be addressed.

Major

  1. The author only focused on the increase of liver and biliary enzyme, however, in clinical practice, the impact on liver function including changes of Alb, PT, ALBI score, and Child-Pugh score is more important than the increase of liver and biliary enzyme. The author should analyze the changes of ALBI score and Child-Pugh score at 12 weeks after PBT. Even though liver function is affected by tumor progression and liver cirrhosis, it is necessary to validate the changes of liver function after PBT. With these results, we can make better therapeutic strategies for HCC.

  1. The author investigated median clinical target volume, but there was no information about target location. Were there any relationship between increase of liver/ biliary enzyme and tumor location ?

  1. The author described 44 patients had alcoholic hepatitis. Alcoholic hepatitis is not same as alcoholic liver disease and the author should confirm the 44 patients fulfilled the definition of alcoholic hepatitis.

  1. The author would describe the information about HCC therapy within 3 month after PBT. Did all patients receive no other HCC therapy within 3 month after PBT ?

Author Response

1, The author only focused on the increase of liver and biliary enzyme, however, in clinical practice, the impact on liver function including changes of Alb, PT, ALBI score, and Child-Pugh score is more important than the increase of liver and biliary enzyme. The author should analyze the changes of ALBI score and Child-Pugh score at 12 weeks after PBT. Even though liver function is affected by tumor progression and liver cirrhosis, it is necessary to validate the changes of liver function after PBT. With these results, we can make better therapeutic strategies for HCC.

→Of course, we consider Child-Pugh is more important than liver enzyme. There are a lot of studies analyzed liver function during radiotherapy. So, we focused on liver enzyme in this study. According to your suggestion, we added change of Child-Pugh score and ALBI in the result section as follows (line number: 110-114):

“Pretreatment and post treatment Child-Pugh score and ALBI score were also investigated. The median Child-Pugh scores were 5 (range: 5 to 10) at pretreatment period and 6 (range: 5 to 10) at the end of the treatment. The median ALBI scores were -2.43 (range: -3.50 to -0.73) at pretreatment period and -2.42 (range: -3.36 to 0.04) at the end of the treatment, respectively. There were no significant changes between pre- and post-treatment. (p=0.511 and p=0.199, respectively)”

2, The author investigated median clinical target volume, but there was no information about target location. Were there any relationship between increase of liver/ biliary enzyme and tumor location ?

→In our institution, treatment protocol is decided only by tumor location written in Materials and Methods section as follows (line number: 184-187): 74 Gy (RBE) in 37 fractions for tumors within 2 cm of the gastrointestinal tract, 72.6 Gy (RBE) in 22 fractions for tumors within 2 cm of the porta hepatis, and 66 Gy (RBE) in 10 fractions for all other tumors. We added p-values in Table 2 to evaluate relationship between increase of liver/ biliary enzyme and tumor location. (line number: 93)

 3, The author described 44 patients had alcoholic hepatitis. Alcoholic hepatitis is not same as alcoholic liver disease and the author should confirm the 44 patients fulfilled the definition of alcoholic hepatitis.

→Thank you for your suggestion. The patients were clinically diagnosed with their habit of drinking, abnormal liver enzymes imaging test, and no virus. We corrected “alcoholic hepatitis” to “alcoholic liver cirrhosis”. (column number: 51 and 57 (Table 1))

4, The author would describe the information about HCC therapy within 3 month after PBT. Did all patients receive no other HCC therapy within 3 month after PBT ?

→No, they received no additional therapy for HCC within 3 months after PBT. We have added additional sentence in the material and method section (line number: 55-56) as follows: “All patients received no additional therapy for HCC within 3 months after completion of PBT.”

Reviewer 2 Report

This study aims at ascertaining liver damage caused by proton beam therapy (PBT) applied to cure hepatocellular carcinoma not treatable with alternative therapeutics. The study employs the pre- and post-treatment levels of hepatic and biliary enzymes and of bilirubin as markers of liver damage. The statistical sample comprises 300 patients with both normal and abnormal pretreatment marker levels, sorted into three groups of ionizing radiation doses/fractions, and followed for up to 3 months after treatment. The study’s main conclusion is that the level of the markers remains almost stable during the whole treatment. This suggests that direct damage of healthy liver cells induced by PBT is minimal even in patients with abnormal pretreatment levels of markers. This may recommend PBT as safe treatment of choice for hepatocellular carcinoma not treatable with alternative approaches. However, these results are incipient, so further and more comprehensive studies may be necessary to finally validate the presented conclusions. Notwithstanding, this study is valuable.

Scientific background, clinical samples, methodologies and procedures are satisfactorily discussed in a concise way. Tables, graphs and color pictures are presented satisfactorily too.  I think that this manuscript will be valuable for the Cancers readership and I would recommend to publish it in its present form.

Author Response

Thanks for your kind comments. We revised our manuscript according to other reviewer`s suggestion.

Reviewer 3 Report

The authors performed a retrospective review on the changes of liver enzymes which are surrogates of liver toxicity after proton beam therapy for hepatocellular carcinoma. The definite strengths of this study is the large number of patients included. That said, the results are expected and only a small number of patients had a transient elevation of liver enzymes which are not clinically significant. The major drawbacks of this retrospective study are addressed below for further consideration by the authors.

(1) The authors did not provide any explanation on why 3 different dose fractionation schemes are used, and how dose/fractionation was chosen; was it based on patient preference, oncologist preference, tumour size, pretreatment liver function or insurance/reimbursement issues? This may have an impact on treatment-related toxicities.

(2) The authors did not perform any logistic regression with univariate and multivariate analyses to identify predictive factors of liver toxicities.

(3) Child-pugh scoring system is the most commonly used tool to assess the liver function. Unfortunately the authors did not perform analysis on the serial changes of Child-Pugh scores before and after proton beam therapy for their patients.

(4) The quality and resolution of the figures can be further improved. The current figures are quite difficult to read and understand.

Thank you very much for your kind invitation.

Author Response

<Answers for reviewer 3>

Comments and Suggestions for Authors

The authors performed a retrospective review on the changes of liver enzymes which are surrogates of liver toxicity after proton beam therapy for hepatocellular carcinoma. The definite strengths of this study is the large number of patients included. That said, the results are expected and only a small number of patients had a transient elevation of liver enzymes which are not clinically significant. The major drawbacks of this retrospective study are addressed below for further consideration by the authors.

(1) The authors did not provide any explanation on why 3 different dose fractionation schemes are used, and how dose/fractionation was chosen; was it based on patient preference, oncologist preference, tumour size, pretreatment liver function or insurance/reimbursement issues? This may have an impact on treatment-related toxicities.

→The 3 fractionations were basically selected only by tumor location as mentioned in “4.2. proton beam therapy” section, line number: 191-194. We would like to select hypofractionation of 66Gy(RBE) in 10 fractions, however, we consider this fractionation had more risk of bile duct stenosis and gastrointestinal ulceration for patients with tumor at hepatic portal or adjacent to alimentary tract, and reduce fraction doses for these patients.According to our results, there is no significant difference in change of liver/biliary enzyme among 3 fractionations. We have added p-value in Table2. (line number: 93)

(2) The authors did not perform any logistic regression with univariate and multivariate analyses to identify predictive factors of liver toxicities.

→In this study, any elevations in the patients with normal pretreatment values were not found (all significant changes were declines from pretreatment values), so we did not perform any more analysis.

(3) Child-pugh scoring system is the most commonly used tool to assess the liver function. Unfortunately the authors did not perform analysis on the serial changes of Child-Pugh scores before and after proton beam therapy for their patients.

→There are a lot of studies analyzed liver function during radiotherapy. So, we focused on liver enzyme in this study. According to your suggestion, we added change of Child-Pugh score and ALBI in the result section as follows (line number: 110-114):

“Pretreatment and post treatment Child-Pugh score and ALBI score were also investigated. The median Child-Pugh scores were 5 (range: 5 to 10) at pretreatment period and 6 (range: 5 to 10) at the end of the treatment. The median ALBI scores were -2.43 (range: -3.50 to -0.73) at pretreatment period and -2.42 (range: -3.36 to 0.04) at the end of the treatment, respectively. There were no significant changes between pre- and post-treatment. (p=0.511 and p=0.199, respectively)”

(4) The quality and resolution of the figures can be further improved. The current figures are quite difficult to read and understand.

→We apologies low quality of the figure. It is maximal resolution of our planning machine. We have changes font style of the figures, and partitioned and enlarged Figure 4 (Figure 4A-4C, line number: 101-109).

Round 2

Reviewer 1 Report

The author sincerely responded to the reviewers' comments. The manuscript is worth publishing in Cancers.

Author Response

The author sincerely responded to the reviewers' comments. The manuscript is worth publishing in Cancers.

→Thank you very much for providing important comments. We are thankful for the time and energy you expended.

Reviewer 3 Report

The authors still could not provide clear explanation and elaboration on how the dose/fractionation is selected for each patient and there is no reference to support their decision making.

Overall I do not think the manuscript will convey new message and change in clinical practice.

Author Response

The authors still could not provide clear explanation and elaboration on how the dose/fractionation is selected for each patient and there is no reference to support their decision making.

Overall I do not think the manuscript will convey new message and change in clinical practice.

→Thank you for pointing out. To clear how to select dose/fraction, we added following sentences (Line 192-201) and added reference.

“We selected the fractionations by the tumor locations. When the tumors were adjacent to high-risk organs, the fraction doses were reduced to avoid severe chronic adverse effects: 2Gy (RBE) per fraction for the tumors near the gastrointestinal tract to prevent gastrointestinal hemorrhage, 3.3Gy (RBE) per fraction for the tumors near the porta hepatis to prevent severe bleeding and biliary tract obstruction, and 6.6Gy (RBE) for the others. Then, treatment schedules were established as 74 Gy (RBE) in 37 fractions for the tumors within 2 cm of the gastrointestinal tract, 72.6 Gy (RBE) in 22 fractions for the tumors within 2 cm of the porta hepatis, and 66 Gy (RBE) in 10 fractions for all the other tumors in our institution. The gastrointestinal tract was avoided as far as possible after 40 to 50 Gy (RBE) irradiated. Previous study has suggested that these three fractionations can achieve similar therapeutic effects.[13]”

The information during radiotherapy for HCC is very limited. So, we believe that our manuscript will be useful for clinicians to perform radiotherapy for HCC.
